# Prevention of Secondary Lymphedema after Complete Lymph Node Dissection in Melanoma Patients: The Role of Preventive Multiple Lymphatic–Venous Anastomosis in Observational Era

**DOI:** 10.3390/medicina58010117

**Published:** 2022-01-13

**Authors:** Eleonora Nacchiero, Michele Maruccia, Fabio Robusto, Rossella Elia, Alessio Di Cosmo, Giuseppe Giudice

**Affiliations:** 1Department of Plastic Reconstructive and Aesthetic Surgery, Università degli Studi di Bari, 70121 Bari, Italy; marucciam@gmail.com (M.M.); rossellaelia4@gmail.com (R.E.); alessiodicosmo@gmail.com (A.D.C.); giuseppe.giudice@uniba.it (G.G.); 2Medonline-Statte, Asl Ta, 74010 Statte, TA, Italy; fabiorobusto@libero.it

**Keywords:** secondary lymphedema, multiple lymphatic venous anastomosis, melanoma, complete lymph node dissection, observational era

## Abstract

*Background and Objectives*: Current guidelines have limited the performance of complete lymph node dissection (CLND) for patients with clinically detectable lymphatic metastases. Despite the limitations of this surgical procedure, secondary lymphedema (SL) is an unsolved problem that affects approximately 20% of patients undergoing CLND. Preventive lymphatic–venous micro-anastomoses (PMLVA) has already demonstrated its efficacy in the prevention of SL in melanoma patients with a positive sentinel lymph node biopsy (SLNB), but the efficacy of this procedure is not demonstrated in patients with clinically detectable lymphatic metastases. *Materials and Methods*: This retrospective cohort study, was performed in two observation periods. Until March 2018, CLND was proposed to all subjects with positive-SLNB andPMLVA was performed in a subgroup of patients with risk factors for SL (Group 1). From April 2018, according to the modification of melanoma guidelines, all patients with detectable metastatic lymph nodes underwent PMLVA during CLND (Group 2). The frequency of lymphedema in subjects undergoing PMLVA was compared with the control group. *Results*: Database evaluation revealed 172 patients with melanoma of the trunk with follow-up information for at least 6 mounts. Twenty-three patients underwent PMLVA during CLND until March 2018, 29 from April 2018, and 120 subjects underwent CLND without any preventive surgery (control Group). The frequency of SL was significantly lower in both Group 1 (4.3% vs. 24.2%, *p* = 0.03) and Group 2 (3.5%, *p* = 0.01). Patients undergoing PMLVA showed a similar recurrence-free periods and overall survival when compared to the control group. *Conclusions*: PMLVA significantly reduces the frequency of SL both in immediate and delayed CLND. This procedure is safe and does not lead to an increase in length of hospitalization.

## 1. Introduction

The incidence of melanoma is constantly increasing across the developed countries, while its survival rates have greatly improved over the last three decades. Currently, the overall 5-year survival rate is 90% in melanoma patients, ranging from 40 to 78% in patients with lymphatic involvement [1]. Sentinel lymph node biopsy (SLNB) is an indispensable procedure in the staging of melanoma [2], but the benefit of this procedure regarding survival is not unequivocal [3]. Complete lymph node dissection (CLND) for patients with positive sentinel lymph nodes (PSLNs) was the standard of care until very recently [4]; at present, recent trials have testified the absence of any impact on survival for immediate CLND, advising periodic follow-up with an ultrasound of the PSLN basin [5,6]. Although CLND does not present a relevant systemic toxicity [7], this surgical procedure is no longer recommended in all patients with positive-SLNs because of the risk of acute and chronic local complications [8,9,10]. To date, CLND is only indicated in the case of isolated locoregional, clinically detectable lymphatic metastases [4].

Lymphedema is the most frequent complication related to CLND at long-term follow-up, affecting up to 20% of patients [11]. Chronic lymphedema is associated with esthetical and functional concerns, and could lead to several complications, such as erysipelas, warts, and papillomatosis cutis lymphostatica [12]. Long-term improvements in lymphedema symptoms are rarely achieved despite the use of surgical or physiotherapeutic therapy. Previous studies have testified the effectiveness of preventive multiple lymphatic-venous anastomosis (PMLVA) during CLND in the prevention of secondary lymphedema (SL) [13]. As melanoma guidelines have been modified, recommending that CLND is only performed in subjects with a clinically detectable lymphatic metastasis, melanoma patients undergoing CLND frequently report a more extensive lymphatic involvement. In this setting, namely, in case of more invasive surgery, the efficacy of PMLVAs in the prevention of SL has not been investigated to date.

## 2. Methods

From a single institution digital database, we identified all patients affected by CM of the trunk who underwent SLNB in the axilla or groin from July 1994 to March 2021. The exclusion criteria are as follows: the presence of stage IV melanoma or other carcinomas; compromised lymphatic drainage secondary to prior procedures. All patients underwent lymphoscintigraphy; technetium–nanocolloid human serum albumin was injected close to the primary lesion. The reduction in artefacts was ensured by the use of ultra-high-resolution collimators [14,15]. Dual-headed digital gamma cameras were used to acquire dynamic and static images after the radiolabeled colloid injection, and then after every node was visualized [16]. Specimens from multiple sections were stained with conventional hematoxylin/eosin and immunohistochemical analysis was performed for each SLN. 

SLNB was performed in the case of melanoma with thickness ≥ 0.75 mm or in case of adverse prognostic features [17]. Until March 2018, CLND was suggested to all patients with positive SLNB; later, this surgical procedure was indicated only in the case of clinically detectable lymphatic metastases, as suggested by new guidelines. Complete therapeutic lymph node dissection involved the three lymphatic levels in the axilla while, in the groin, this included the iliac and obturator lymph nodes [18]. PMLVAs were performed during CLND in a subgroup of patients undergoing lymphadenectomy before March 2018 and in all patients at risk for SL thereafter. The following risk factors for the development of lymphedema were considered: (i) previous radiotherapy, (ii) complicated surgical wounds or seroma, (iii) obesity, (iv) infections, (v) chronic inflammatory cutaneous disorders, (vi)vascular hypertension, (vii) congenital predisposition, (viii) traumatism, (ix) chronic kidney disease, cardiac failure, and malnutrition, (x) arteriovenous shunts, pacemaker, implants, (xi) orthopedic surgery, (xii) venous insufficiency or postphlebitic syndrome, thrombophlebitis, (xiii) familiar anamnesis for chronic edema, and (xiv) hyperthyroidism. 

Patent Blue Violet was injected intradermically and at a deep level to prepare for PMA; a “T” shape was then incised into the skin. PMAs were preferably performed under the same surgical access as CLND; when this was not feasible, a new incision was made at the middle third of the involved limb. Under a microscope magnification of 25×, lymphatic vessels were identified by Patent Blue Violet dye staining and isolated. After the identification and isolation of a vein with sufficient flow, its distal portion was ligated while its proximal portion was clamped. A Nylon 11/0 was used to transfix lymphatic vessels and suture them to the vein through end-to-end telescopic anastomosis with a “Donati” stitch. At this point, single stitches were affixed between the lymphatic and the vein wall. Finally, the first stich and the clamp were removed.

All patients were followed-up in an outpatient setting at 6-month intervals for the first 5 years, and then once a year. They were asked to perform a chest X-ray, and abdominal and lymphatic ultrasound. SPECT/CT was indicated for difficult areas. During each follow-up visit, we measured calf and bicep circumference and water volumetry [19]. The criteria to define lymphedema were: (1) an increase of 7% or more in the sum of circumferences of the defined points along the limb; (2) an increase of 15% or more of the whole limb volume [20,21]. Moreover, body mass index (BMI) was registered during each follow-up visit. A BMI value up to 30 was used as criteria to define obesity and was considered a risk factor for developing lymphedema.

In Group 1, we included the patients who underwent PMLVAs during CLND before April 2018, while patients in whom this preventive technique was used later were included in Group 2. The difference between these groups consisted in the indication of CLND: -Group 1: Patients underwent CLND in any case of report of positive-SLNB;-Group 2: Patients underwent CLND only in the case of clinical detectable lymphatic metastasis.

All subjects in whom no preventive surgical technique was performed were included in the control group.

We received approval from the ethics committee of our institution for the implementation of this research protocol (approval code 5029, approval date 18 October 2017). Written informed consent was obtained from all the patients before the surgical procedure. Statistical analysis was performed with *t*-test, Chi-square, and Wilcoxon rank sum test as appropriate. Follow-up time was defined as the time between definitive surgical treatment of the primary melanoma and last contact with the patient. Patients with less than 3 months of follow-up were excluded from the study. Survival rates were estimated using the Kaplan–Meier method and multivariate Cox proportional hazards regression models. *p* values < 0.05 were considered statistically significant. All statistical analyses were performed using SAS Software Release 9.4 (SAS Institute, Cary, NC, USA).

## 3. Results

Overall, CLND was performed in 172 patients with melanoma of the trunk and a minimum follow-up of 6 mounts. One-hundred-and-forty-three underwent lymphadenectomy before March 2018 in case of positive SLNB. PMLVA was performed in the same surgical time as CLND in 23 subjects (Group 1), while, in the remaining 120 patients, no preventive surgery was adopted (control group). Otherwise, in accordance with the application of the new guidelines from April 2018, CLND was indicated in 29 subjects and performed only in the case of isolated, locoregional, clinically detectable lymphatic metastases (Group 2). Groups were homogeneous for age, sex, BMI, primary tumor ulceration, mitotic rate, and Breslow thickness (Table 1).

Additional non-sentinel lymph nodes were detected in 32 (26.7%) subjects in the control group, in 7 (30.4%) subjects in Group 1 and in 7 subjects in Group 2 (*p* = 0.88). During follow-up, 55 (45.8%) deaths were recorded in the control group, in 9 (39.1%) subjects in Group 1 and in 10 (34.5%) subjects in Group 2 (*p* = 0.50), while, 67 (55.8%) patients in the control group, 11 (47.8%) in Group 1, and 14 (48.3%) in Group 2 developed recurrences. No significant differences in both overall survival (log-rank *p* = 0.72) and disease-free period (log-rank *p* = 0.69) were detected among the three groups using Kaplan–Meier analysis at 3 years of follow-up (Figure 1 and Figure 2).

Overall, SL was detected in 31 of 172 (18.0%) subjects undergoing CLND. Upper-limb lymphedema after axillary dissection is less frequent than lower-limb lymphedema after groin dissection (10.0% vs. 23.5%, Chi-square *p* = 0.02). Considering body mass index (BMI), we detected 31 (18.0%) obese subjects; there was an insignificantly (*p* = 0.09) higher frequency of obesity among subjects who developed lymphedema (29.6%) than in the rest of the population (15.8%). The use of PMLVA led to a significant reduction in secondary lymphedema in both Group 1 (*p* = 0.03) and in Group 2 (*p* = 0.01) when compared with subjects included in the control group. In fact, frequency of SL was 24.2% (29 of 120 subjects) in the control group, 4.3% (1 of 23) in Group 1, and 3.5% (1 of 29) in Group 2. PMLVA did not appear to be differently beneficial for the axillary or inguinal district (Table 2). A multivariate logistic regression was conducted to study the risk factors for the development of lymphedema (age, sex, obesity, basin site, and execution of PMLVA). PMLVA resulted an independent protective factor for SL in both Group 1 (OR 0.123, 95% CI 0.015–0.980, *p* < 0.05) and Group 2 (OR 0.319, 95% CI 0.114–0.890, *p* < 0.05) (Table 3).

The performance of PMLVA significantly increased the surgical time; in fact, the mean duration of surgical procedure in Group 1 and Group 2 was 1.70 ± 1.49 h and 1.69 ± 0.56 h, respectively; this was significantly higher when compared to the 0.51 ± 0.17 h of the control group (*p* < 0.001). This significantly longer operating time between subjects undergoing PMLVA and controls was confirmed in both axillary CLNB and groin CLNB. In fact, axillary dissections required a mean time of 0.55 ± 0.21 h in the control group, 1.61 ± 0.48 h in Group 1, and 1.82 ± 0.57 h in Group 2, while the mean duration for groin dissection was 0.49 ± 0.14 h in the control group, 1.73 ± 0.51 h in Group 1 and 1.61 ± 0.56 h in Group 2. On the contrary, the mean length of stay appeared to be similar among the groups: 10.32 ± 1.12 days in the control group, 10.36 ± 1.56 days in Group 1 and 10.00 ± 1.81 days in Group 2 (*p* = 0.90). 

## 4. Discussion

CLND is reserved for melanoma patients with clinically detectable lymphatic metastases, reducing the number of subjects that undergo lymphatic dissection. In prophylactic era, there was a high frequency of CLND in patients with no metastatic lymph nodes. If, on the one hand, the modifications to CLND indications led to a reduction in the number of patients exposed to complications without an improvement in prognosis [7], then currently, patients in whom lymphatic dissections were performed presented a more advanced stage of lymphatic disease. Moreover, the need to perform as exhaustive a CLND as possible was reaffirmed by the adoption of a number of excised lymph nodes as a quality assurance measure for CLND [22]. Regrettably, the execution of more radical CLND led to an increased frequency of local complications, particularly lymphedema [23]. Our report shows an SL frequency of up to 20% of melanoma subjects undergoing CLND, mimicking the findings reported in recent studies [24]. Over the years, many surgical approaches have been proposed to reduce the risk of SL, including minimally invasive laparoscopic surgery or lymph node transfers [25,26,27,28,29,30,31,32,33,34,35,36,37]. Furthermore, compressions or massages are usually used to limit or prevent secondary lymphedema in post-operative rehabilitation [38].

The efficacy of PMLVA was already testified for the prevention of secondary lymphedema after CLND of the axilla or groin in oncologic patients [39]. Moreover, our previous study—whose data have been included in the current article—have demonstrated the efficacy of PMLVA in reducing the risk of SL in melanoma patients exposed to CLND; however, this analysis included all patients with a positive SLNB. 

The present analysis has testified the efficacy of the execution of PMLVA during node dissection in reducing the risk of SL in patients who underwent delayed CLND after a clinical evidence of nodal metastasis; therefore, there is a more advanced metastatic involvement of the lymphatic basin than was seen in the previous cohort.

The results of the current study have testified a similar improvement in the prevention of SL in both Group 1 (patients undergoing CLND after a positive SLNB) and Group 2 (patients with clinically detectable lymphatic metastases) with respect to the control group. The efficacy of PMLVA in the reduction in the incidence of SL has been demonstrated even when the results were corrected by other risk factors in a multivariate analysis. This evidence suggests the utility of PMLVA in the prevention of SL, as well as in the case of more advanced lymphatic metastasis, requiring a more invasive surgery.

Moreover, our data have identified that the execution of PMLVA did not impact the prognosis of melanoma patients; in fact, overall-survival and recurrence-free period were similar among groups. Another point in favor of PMLVA is the absence of differences in length of stay among the groups. On the contrary, an increased duration of surgery was reported in patients undergoing PMLVA. Although, on the one hand, the increased surgical time could require more staff and increased instrumental resources, on the other hand, the proven effectiveness of PMLVA in preventing SL could generate cost savings for surgical or physiotherapeutic interventions to treat SL. Thus, the benefits of PMLVA involve lower direct and indirect costs related to SL.

## 5. Conclusions

This report has demonstrated the efficacy of PLMVA in the prevention of secondary lymphedema in melanoma patients in the observational era, as well as testifying to its effectiveness in subjects with clinical detectable metastasis and a more advanced lymphatic involvement. This surgical technique was safe and had no impact on the length of hospitalization. PMLVA could also be used for the prevention of SL after SLNB; in fact, although this procedure is less commonly associated with SL, the literature reports an incidence of almost 2% in melanoma patients [40].

## Figures and Tables

**Figure 1 medicina-58-00117-f001:**
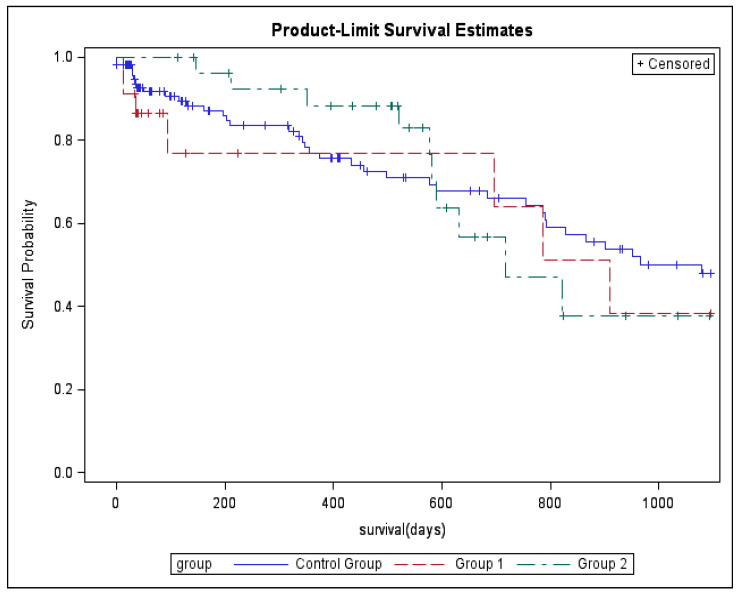
Kaplan–Meier curves for the cumulative risk of death at 3 years.

**Figure 2 medicina-58-00117-f002:**
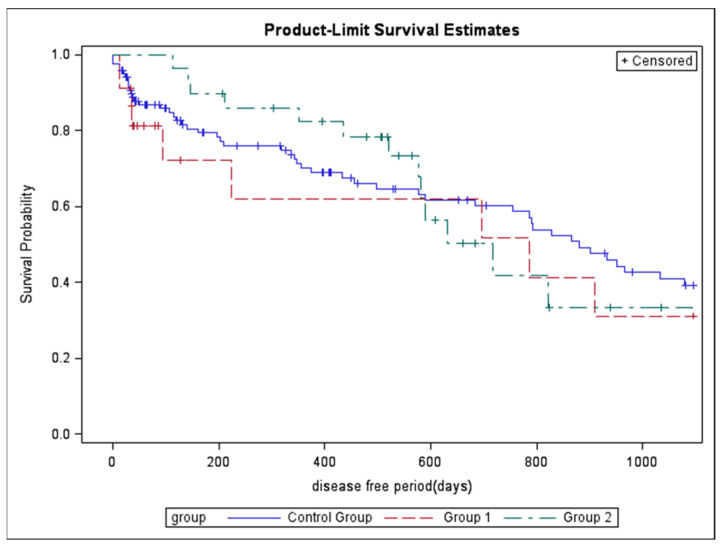
Kaplan–Meier curves for the cumulative risk of recurrence at 3 years.

**Table 1 medicina-58-00117-t001:** Characteristics of subjects that underwent CLND.

Characteristics	Control Group	Group 1	Group 2	*p*
Total, n (%)	120	23	29	/
Gender (M), n (%)	64 (53.3)	12 (52.2)	14 (48.3)	0.89
Age (y)				
Median	53.3	54.8	54.2	0.43
Range	18.9–74.2	18.0–68.9	19.8–81.1	
Breslow thickness, n (%)				
<1 mm	24 (20.0)	5 (21.7)	5 (17.2)	0.99
1–2 mm	53 (44.2)	10 (43.5)	11 (37.9)	
2–4 mm	26 (21.7)	5 (21.7)	8 (27.6)	
>4 mm	17 (14.2)	3 (13.0)	5 (17.2)	
Ulceration, n (%)	27 (22.5)	5 (21.7)	8 (27.6)	0.83
>1 mitosis/mm, n(%)	17 (14.2)	5 (21.7)	7 (24.1)	0.35
Obesity (BMI > 30 kg/m^2^)	16 (13.3)	5 (21.7)	6 (20.7)	0.43
Duration of surgical procedure (hours)				
Median	0.50	1.75	2.00	<0.01
Range	0.25–1.00	0.50–2.50	0.75–2.75	
Length of stay (days)				
Median	10	10	10	0.90
Range	8–14	8–14	8–14	

**Table 2 medicina-58-00117-t002:** Frequency of lymphedema. Comparison with the results of group 1, group 2 and the control group.

CLND Site	Control Group	Group 1		Group 2	
	*N*	*N*	*p*	*N*	*p*
Axilla	7/51 (13.7%)	0/8 (0.0%)	/	0/11 (0.0%)	/
Groin	22/69 (31.9%)	1/15 (7.7%)	0.05	1/18 (5.6%)	0.02
TOT	29/120 (24.2%)	1/23 (4.3%)	0.03	1/29 (3.5%)	0.01

**Table 3 medicina-58-00117-t003:** Multivariate Cox proportional hazards regression models. Comparison among the groups.

Characteristics	Group 1		Group 2	
	HR	95% CI	HR	95% CI
Age	1.001	0.982 1.019	1.006	0.989 1.013
Sex (M)	1.503	0.603 3.743	1.180	0.477 2.919
Obesity	1.738	0.588 5.138	2.028	0.673 6.115
Lymphatic Basin (groin)	2.542	0.952 6.787	2.796	1.051 7.440
PMLVA	0.123	0.015 0.980	0.319	0.114 0.890

## Data Availability

The data presented in this study are available on request from the corresponding author. The data are not publicly available due to privacy and ethical reasons.

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
