# Peer review of "Prevention of Secondary Lymphedema after Complete Lymph Node Dissection in Melanoma Patients: The Role of Preventive Multiple Lymphatic–Venous Anastomosis in Observational Era"

_medicina, 2022, doi:10.3390/medicina58010117_

Round 1

Reviewer 1 Report

Insert the paragraph Conclusions in the text

Author Response

The paragraph “conclusions” has been added in the manuscript.

Reviewer 2 Report

The authors described "Prevention of secondary lymphedema after complete lymph node dissection in melanoma patients: the role of preventive multiple lymphatic-venous anastomosis in observational era." As they pointed out,  previous studies have testified the effectiveness of preventive multiple lympho-venous anastomosis during CLND in the prevention of secondary lymphedema. In addition, there have been no report whether PMLVA can reduce the frequency of SL both in immediate and delayed. However, I have a concern about the safety of LVA in patients with cancer.

The authors described that patients undergoing PMLVA showed similar recurrence-free periods and overall survival when compared to the control group. But, previous animal study set off alarm bells throught out the surgery. Especially in the clinical situation, a high degree of caution is advised when “malignancy” exists (Hayashida K, et al. Plast Reconstr Surg, 2017). Please more expand the safety of the surgery citing previous literature in Discussion.

Author Response

The authors described "Prevention of secondary lymphedema after complete lymph node dissection in melanoma patients: the role of preventive multiple lymphatic-venous anastomosis in observational era." As they pointed out, previous studies have testified the effectiveness of preventive multiple lympho-venous anastomosis during CLND in the prevention of secondary lymphedema. In addition, there have been no report whether PMLVA can reduce the frequency of SL both in immediate and delayed. However, I have a concern about the safety of LVA in patients with cancer.

Reply: We understand the reviewer’s concern about the safety of LVA after surgery for malignancy. However, patients with a malignancy involving an extremity (e.g. cutaneous melanoma) are ineligible for this surgery due to oncologic concerns of facilitating metastatic spread of the disease. Conversely, PMLVA has been described for the prevention of extremity lymphedema in patients with melanoma of the trunk and vulvar cancer and has been reported to be a feasible and safe procedure in this scenario.

The authors described that patients undergoing PMLVA showed similar recurrence-free periods and overall survival when compared to the control group. But, previous animal study set off alarm bells throught out the surgery. Especially in the clinical situation, a high degree of caution is advised when “malignancy” exists (Hayashida K, et al. Plast Reconstr Surg, 2017). Please more expand the safety of the surgery citing previous literature in Discussion.

Reply: The Discussion has been updated including the reference to previous animal studies. Herein, the added paragraph: “The scenario of a more advanced disease could arise some concerns about the safety of the surgical procedure as previous animal models [39] have advised a high degree of caution when “malignancy” exists. Nevertheless, although mouse models are widely used in lymphatic research due to their accessibility, cost-effectiveness, anatomical and histological representation of human lymphedema, a potential pitfall of the mouse hind limb model is that a chronic lymphedematous state is often elusive. Mouse models exhibiting short lymphedematous states with spontaneous resolution may confound therapies intended to target the chronic lymphedematous nature in humans.  Moreover, a proper comparable clinical setting with the presented human one (malignancy involving the trunk and PMLVA) has not been reported in literature”

Reviewer 3 Report

Throughout the document there are grammatical errors and errors in the layout.

You use a lot of abbreviations, this makes it difficult to read. Also there are some inconsistencies in the use of these abbreviations. Such as LS/SL; PMA?; PMLA/PMLVA

What is the difference with your previous work? The aim of this study is not clear. line 14-15; 52-53

Introduction

line 40: what do you mean by PSLN basin?

line 46: you refer to your one reference when talking about the incidence of lymphedema, you should better use another reference like systematic review.

Methods

Please rewrite the sentence  line 57

Is the information on line 58-64 relevant?

Line 72: How do you define patients at risk for development of lymphedema?

line 93-94: the criteria to define lymphedema, why didn't you made the choice to use 10% relative arm volume difference? Is this compared to the other arm?

line 95: old references, please use recent ones

You don't describe the different groups in your methods. It appears that groin and axilla interventions are included? Please clarify your inclusion criteria.

Results

please refrase line 113

can you at some structure in the results section

Discussion

Refrase line 158-160

Line 166: there is no evidence that preventive manual lymph drainage is efficient

What is the relevance of references 25-37?

Please discuss the difference with your previous work

line 184: identified instead of testified

There is no conclusion?

Disclosures?

Acknowledgements?

Author Response

You use a lot of abbreviations, this makes it difficult to read. Also there are some inconsistencies in the use of these abbreviations. Such as LS/SL; PMA?; PMLA/PMLVA

Reply: We have corrected the inconsistencies in the use of these abbreviations.

What is the difference with your previous work? The aim of this study is not clear. line 14-15; 52-53

Reply: This work is an update of the previous one. Since the previous study, melanoma guidelines have changed the indication for CLND reserving the procedure only to patients with clinically detectable metastases. CLND is now performed in patients with a higher stage of metastatic lymph node involvement. Efficacy of preventive multiple lymphatic-venous anastomosis in a more destructive surgery was not taken for granted. Both the abstract and the introduction have been revised in order to make more clear the original aim of the paper.

Introduction

line 40: what do you mean by PSLN basin?

Reply: We mean the lymphatic basin in which was detected a positive sentinel lymph node.

line 46: you refer to your one reference when talking about the incidence of lymphedema, you should better use another reference like systematic review.

Reply: The reference for the incidence of lymphedema has been changed.

Methods

Please rewrite the sentence line 57

Reply: the sentence starting at line 57 has been rewritten as follows: “As melanoma guidelines have been modified recommending to perfom of CLND only in subjects with a clinically detectable lymphatic metastasis, melanoma patients undergoing CLND frequently report a more extensive lymphatic involvement. In this setting, namely in case of more invasive surgery, the efficacy of PMLVAs in prevention of SL has not been investigated so far”.

Is the information on line 58-64 relevant?

Reply: We think that this information is useful to clarify methodology.

Line 72: How do you define patients at risk for development of lymphedema?

Reply: Risk factor for developing lymphedema were considered: (i)Previous radiotherapy, (ii) complicated surgical wounds or seroma, (iii) obesity, (iv)infections, (v)chronic inflammatory cutaneous disorders, (vi)vascular hypertension, (vii) congenital predisposition, (viii) traumatism, (ix) chronic kidney disease, cardiac failure, and malnutrition, (x) arteriovenous shunts, pacemaker, implants, (xi)orthopedic surgery, (xii)venous insufficiency or postphlebitic syndrome, thrombophlebitis, (xiii) familiar anamnesis for chronic edema, and (xiv) hyperthyroidism (lines 80-86)

line 93-94: the criteria to define lymphedema, why didn't you made the choice to use 10% relative arm volume difference? Is this compared to the other arm?

Reply: We preferred to consider the increase of 15% or more of the whole limb volume when compared with control limb as suggested by Starritt EC et al. in their analysis based on classification and regression tree analysis.

line 95: old references, please use recent ones

Reply: We reported the citations of Starritt EC at al. and Spillane AJ et al. for the specificity of cut-off in definition of lymphedema using water volumetry.

You don't describe the different groups in your methods. It appears that groin and axilla interventions are included? Please clarify your inclusion criteria

Reply: The inclusion criteria of the groups have been clarified (lines 151-159).

++++Results

please refrase line 113

Reply: Line 113 has been refrased (now line 174)

Discussion

Refrase line 158-160

Reply: The sentence at lines 158-160 was not clear and it did not add relevant data to the discussion. Therefore, we decided not to include it in the revised version of the manuscript

Line 166: there is no evidence that preventive manual lymph drainage is efficient

Reply:  Actually, a systematic review evaluating 8 randomized controlled trials examining a potential preventive effect of exercise on SL incidence showed that exercise in the form of progressive resistance training as well as combined exercise therapies consisting of physiotherapy, physical therapy, MLD, stretching, massage, and/or kinesiotherapy are safe and might have a preventive effect on SL incidence.

(Baumann F, T, Reike A, Hallek M, Wiskemann J, Reimer V: Does Exercise Have a Preventive Effect on Secondary Lymphedema in Breast Cancer Patients Following Local Treatment - A Systematic Review. Breast Care 2018;13:380-385. doi: 10.1159/000487428) This reference has been added in the discussion.

What is the relevance of references 25-37?

Reply: References 25-37 are referred to the cited minimally invasive laparoscopic surgery or lymph node transfers procedures aiming to prevent the onset or the progression of  lymphedema. Those references have been included for the sake of completeness in the discussion section, as alternatives to LVA or PLVA procedures. 

Please discuss the difference with your previous work

Reply: the main difference with the previous work has been better clarified in the introduction section: “As melanoma guidelines have been modified recommending to perfom of CLND only in subjects with a clinically detectable lymphatic metastasis, melanoma patients undergoing CLND frequently report a more extensive lymphatic involvement. In this setting, namely in case of more invasive surgery, the efficacy of PMLVAs in prevention of SL has not been investigated so far”. The discussion has been deepened on this point at lines 276-285.

line 184: identified instead of testified

Reply: The word “identified” has been substituted to “testified” (now line 260)

There is no conclusion? Disclosures? Acknowledgements?

Reply: The Conclusion paragraph has now been added in the manuscript, as well as disclosures and acknowledgments.

Round 2

Reviewer 2 Report

The authors revised the manuscript precisely. Thank you for this opportunity.

This manuscript is a resubmission of an earlier submission. The following is a list of the peer review reports and author responses from that submission.